**Analysis**

# A systematic review and meta-analysis of Zika virus epidemiology

Zika virus (ZIKV), classified as a priority pathogen by the World Health Organization, is an *Aedes*-borne arbovirus that can cause neurological complications and birth defects in newborns of mothers infected during pregnancy. We conducted a systematic review of peer-reviewed studies reporting ZIKV epidemiological parameters, transmission models and outbreaks (PROSPERO CRD42023393345) to characterize its transmissibility, seroprevalence, risk factors, disease sequelae and natural history. We performed meta-analyses of the proportions of congenital Zika syndrome, pregnancy loss among ZIKV-infected mothers and symptomatic cases. We extracted information from 574 studies. Across 418 included studies assigned a high-quality score, we extracted 969 parameters, 127 outbreak records and 154 models. Using random-effects models, we estimated proportions of congenital Zika syndrome (4.65%, 95% confidence interval (CI): 3.38–6.67%), pregnancy loss (2.48%, 95% CI: 1.62–3.78%) and symptomatic cases (51.20%, 95% CI: 38.00–64.23%). Seroprevalence estimates ($n$ = 354) were retrieved beyond South America and French Polynesia. Basic reproduction number estimates ($n$ = 77) ranged between 1.12 and 7.4. We found 66 human epidemiological delay estimates, including the intrinsic incubation period ($n$ = 11, range: 4–12.1 days), infectious period ($n$ = 15, range: 3–50 days), extrinsic incubation period ($n$ = 22, range: 5.1–24.2 days) and serial interval ($n$ = 27, range: 7.4–32.9 days). These data are available in the R package 'epireview' (version 1.4.5). We provide a comprehensive systematic summary of ZIKV epidemiology, revealing large heterogeneities and inconsistencies in the reporting of parameter estimates, study designs and parameter definitions and underscoring the need for standardized epidemiological definitions.

Arboviruses pose a substantial threat to public health, with the World Health Organization (WHO) reporting close to 5.5 billion people at risk globally[1,2]. ZIKV is an arbovirus of concern that is mainly transmitted via *Aedes* mosquitoes but also human-to-human via sexual transmission[3]. Initially discovered in Uganda in 1947, ZIKV was only detected on the African and Asian continents until the mid-2000s[4]. The first outbreak of ZIKV outside these regions occurred in the Federated States of Micronesia in 2007 (ref. 5), with the virus subsequently spreading to the Americas[6,7]. An increase in cases of ZIKV, microcephaly, congenital Zika syndrome (CZS) and Guillain–Barré syndrome in Brazil in 2015 led to the declaration of a Public Health Emergency of International Concern (PHEIC) by the WHO from February to November 2016 (refs. 8–10).

While many ZIKV infections are asymptomatic[3,5,11], the serious complications in the fetus and negative pregnancy outcomes associated with ZIKV infection during pregnancy, including CZS and its association with Guillain–Barré syndrome and other neurological

✉e-mail: k.mccain22@imperial.ac.uk; a.vicco21@imperial.ac.uk; i.dorigatti@imperial.ac.uk

disorders, underscore why future ZIKV re-emergence remains a global public health concern[10]. To date, no licensed vaccines or therapeutics exist, and although ZIKV cases have declined globally since 2017, they continue to be detected worldwide[12]. ZIKV remains listed among the pathogens with pandemic potential and is prioritized for research and development by the WHO[13,14] and the UK Health Security Agency (UKHSA)[15].

Mathematical models play a critical role in outbreak response and risk assessment by providing insights into the potential location, timing and magnitude of disease transmission. These tools can help guide the selection of sites for vaccine or therapeutic trials and inform the prepositioning of protocols and other preparedness activities. Central to these activities is the need for quantitative estimates of key epidemiological parameters, such as the reproduction number, epidemiological delays and seroprevalence, together with their variation and a catalogue of existing mathematical models. Here, we conducted a systematic review to generate a comprehensive database to aid the global modelling and public health community in their response to future ZIKV outbreaks. This systematic review is part of a set of reviews of nine WHO priority pathogens[16–19] and contributes to a wider central, up-to-date and accessible database of information critical to mathematical modelling for timely outbreak response.

## Results

From the 27,491 studies identified through the database search, 11,845 studies were screened following de-duplication, of which 101 were identified through backwards citation screening of the systematic reviews. A total of 10,500 studies were excluded at the abstract/title screening stage, and we undertook full-text review of the remaining 1,343 studies, of which 574 met the criteria for inclusion (Fig. 1 and Supplementary Table A1). We extracted 159 outbreak records, 229 models and 1,334 parameters, of which 127 outbreak records, 154 models and 969 parameters belonged to publications that met the quality score ≥ 50% and are included in our main analysis. All extracted data and the full list of publications included in the main text of this review are available in the R package 'epireview' (version 1.4.5)[20]. The full analysis, including all publications, regardless of their quality assessment score, is provided in Supplementary Figs. B25–B28). Information about the extracted models is provided in Table B7 and Fig. B4.

### Outbreaks

Forty studies reported 127 outbreak records between 2006 (Thailand) and 2021 (India) from 29 countries across the Americas ($n = 97$), Oceania ($n = 13$), Asia ($n = 7$), Africa ($n = 4$) and Europe ($n = 2$; Fig. 2 and Supplementary Table B8). The largest number of records was reported in Colombia across several subregions ($n = 55$), followed by French Polynesia ($n = 12$) and Suriname ($n = 11$). The largest ZIKV outbreaks were reported in Puerto Rico between 2015 and 2016 (with 39,717 suspected cases, of which 36,390 were laboratory-confirmed) and in Colombia between 2015 and 2017 (with 108,087 suspected cases, 62% of which were among women, and 9,802 of which were laboratory-confirmed).

### Parameters

The 969 extracted parameter estimates came from 281 studies, comprising seroprevalence ($n = 354$), risk factors ($n = 242$), reproduction numbers ($n = 117$), CZS proportion ($n = 53$), epidemiological delays in humans ($n = 66$), attack rates ($n = 52$), extrinsic incubation period ($n = 22$), evolutionary rate ($n = 15$), substitution rate ($n = 3$), pregnancy loss proportion ($n = 23$), case fatality ratio (CFR; $n = 6$), proportion of symptomatic cases ($n = 10$), relative contributions to transmission ($n = 2$) and growth rates ($n = 4$) (Supplementary Fig. B3).

### Seroprevalence

A total of 139 studies reported 354 ZIKV seroprevalence estimates from serosurveys conducted between 1980 (Nigeria[21]) and 2022

(El Salvador[22]) in 59 countries across the Americas ($n = 143$), Africa ($n = 97$), Asia ($n = 62$), Europe ($n = 36$) and Oceania ($n = 16$). All seroprevalence estimates extracted in this study are reported in Supplementary Table B10. Almost one-third of these estimates were from surveys of the general population ($n = 107$), while the rest were from pregnant women ($n = 58$), suspected ZIKV infection ($n = 64$), children ($n = 35$), mixed population groups ($n = 35$), blood donors ($n = 10$) and other at-risk populations ($n = 43$) (Supplementary Figs. B9 and B10). The timing of serosurveys coincided with the major epidemics (Supplementary Figs. B9 and B10). The majority of ZIKV seroprevalence estimates were derived from immunoglobulin G (IgG) assays ($n = 139$) and neutralization tests ($n = 109$; Supplementary Fig. B11).

Seroprevalence estimates varied widely from 100% in a hospital-based study of 29 individuals in Cúcuta, Colombia[23], to 0% in local regions in Austria[24], Iran[25] and Taiwan[26], and specific locations within Brazil[27,28], Bolivia[29], Kenya[30], Indonesia[31] and Thailand[32]. Within Brazil, ZIKV exposure in Salvador (Bahia) ranged between 7% (95% CI: 4–10%) in 2014 and 63% (95% CI: 60–65%) in 2015 (ref. 33), consistent with the timing of the major outbreak in the Americas. In contrast, ZIKV seroprevalence in Rio de Janeiro was reported to be 3% in 2018 (ref. 34). In Asia, a national serosurvey conducted in Thailand during 2017–2020 yielded an average seroprevalence of 2.8%[35], but estimates were as high as 23.5% from a serosurvey conducted in 2011–2012 in Nakhon Sawan city[36]. In Africa, serosurveys conducted from 2013 to 2017 reported ZIKV seroprevalence less than 11.5% in several sites of Kenya[30], 42.1% in Gabon[37], 10.9% in Cabo Verde[38] and ranging from 3.1% to 43.3% in Mali[25].

### Transmissibility

We extracted estimates of the basic reproduction number $R_0$ ($n = 77$), human-to-human (sexual) $R_0^h$ ($n = 14$) and vector-borne $R_0^v$ ($n = 9$), effective reproduction number $R_e$ ($n = 15$) and sexual $R_e^h$ ($n = 2$) from 40 studies. The methods used to estimate $R_0$ and $R_e$ varied across the different studies. Compartmental models were used most frequently ($n = 45$), and among these, the next-generation matrix method was used for 27 estimates. Branching processes were used for 16 estimates. Most estimates were from the 2013–2014 outbreaks in French Polynesia ($n = 24$), and the 2015–2016 epidemics in Brazil ($n = 26$) and Colombia ($n = 15$).

The majority (63/77) of the central estimates of $R_0$ were between 1.5 and 4 (Fig. 3). Estimates of the other transmission type-specific reproduction numbers are shown in Supplementary Figs. B5 and B7 and Table B12).

### Epidemiological delays

We analysed 66 human epidemiological delays from 29 studies and 22 extrinsic incubation period estimates from 12 studies (Fig. 4). The 11 central intrinsic incubation period estimates ranged from 4 days (95% CI: 3.15–5.48) in Australes, French Polynesia in 2013–2014 (ref. 39) to 12.1 days in Brazil in 2016 (ref. 40), although roughly half of the reported estimates ($n = 6/11$) were between 5 and 7 days. The 15 estimates of the human infectious period were highly variable, ranging from central estimates of 3 days in Brazil in 2016 (ref. 40) to 50 days among asymptomatic cases in Florida, United States[41] in July to September 2016. We extracted only two estimates from two studies of the serial interval: a mean of 7.4 days (95% CI: 4.59–10.2) in Singapore in August to November 2016 (ref. 42) and of 32.9 days in French Polynesia between October 2013 and October 2016 (ref. 43). The single extracted estimate for the generation time was estimated from data collected in the French West Indies (Guadeloupe, Martinique and Saint-Martin) in 2015–2017 and was described by a mean of 2.5 weeks (s.d. 0.7).

The 22 central estimates of the extrinsic incubation period ranged from 5.1 (ref. 44) to 24.2 days[44,45]. The central estimates of the time from symptom onset to admission to care ranged from 1 (interquartile range (IQR) 1–2) day among hospitalized patients in Colombia[46] to 8 (IQR 4–21) days among travellers[47]. The central estimates of time from admission to recovery or discharge ranged from 4 (ref. 48) to 23

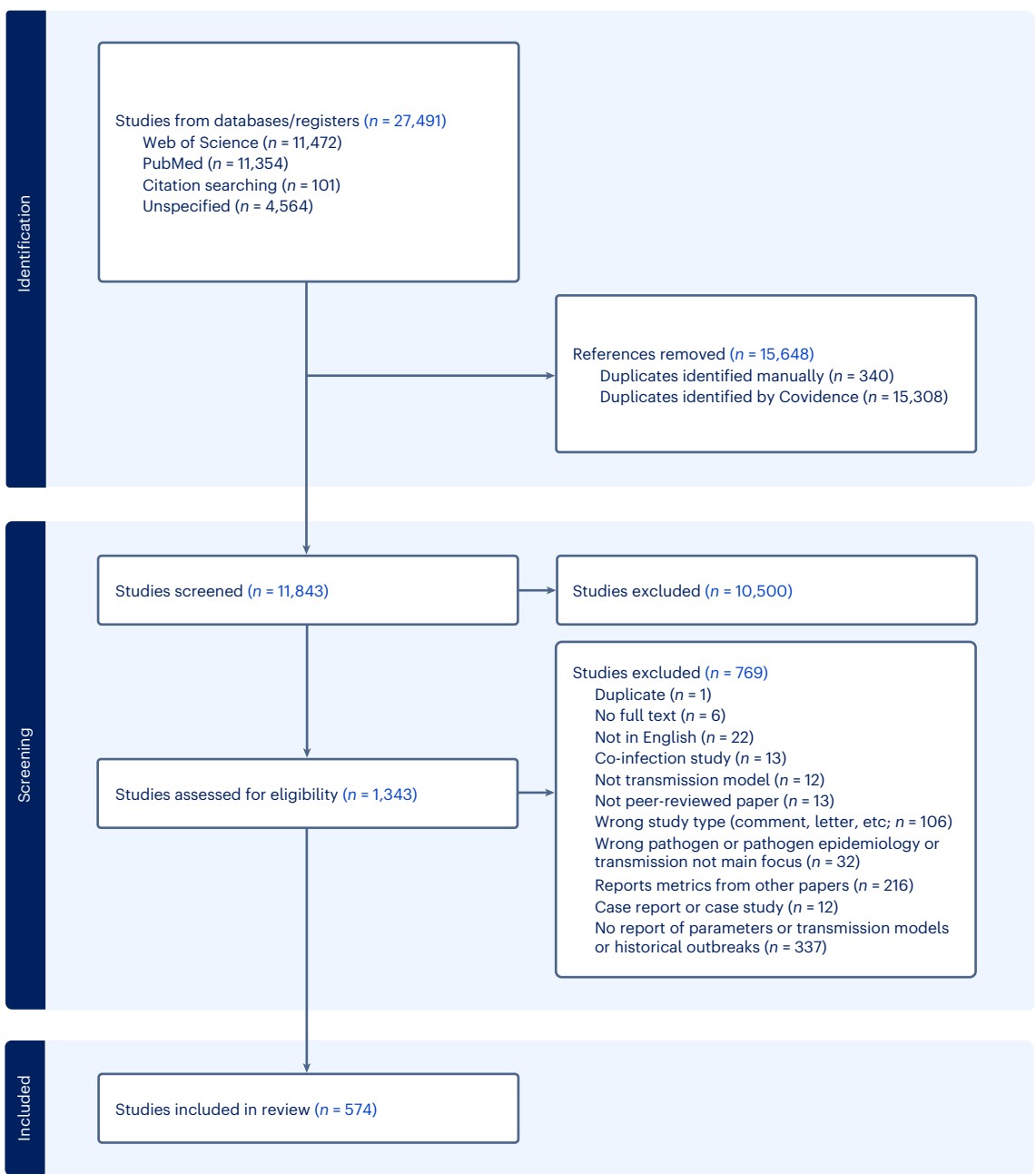

**Fig. 1 | Flow chart for study selection.** Studies were screened according to the Preferred Reporting Items for Systematic Reviews and Meta-Analyses (PRISMA) guidelines (Supplementary Information). Reasons for exclusion at the abstract or title stage were not collected by the Covidence software.

(IQR 9.5–36)[23] days. All epidemiological delays extracted in this study are listed in Supplementary Table B9.

### Severity and adverse birth outcomes

We extracted 53 estimates of CZS proportion given a ZIKV-infected mother from 36 studies. The CZS proportion was highly variable across studies (Fig. 5a), 92.5% (49/53) of estimates were from the Americas, and all estimates were obtained from data collected between January 2015 and February 2020.

Among the 53 estimates, 39 estimates from 24 studies were eligible for meta-analysis based on our stringent meta-analysis inclusion criteria (see 'Data analysis' in the Methods). The pooled CZS proportion, restricted to non-trimester-specific estimates, was 4.65% (95% CI: 3.38–6.37%) using the random-effects model ($I^2 = 77.10\%$; Supplementary Fig. B33). Stratifying the meta-analysis by trimester of maternal ZIKV exposure showed a higher pooled estimate of CZS proportion in the first trimester (6.50%, 95% CI: 1.31–26.51%, $I^2 = 86.10\%$) compared to the second (4.96%, 95% CI: 2.91–8.34%, $I^2 = 57.40\%$) or third (0.71%, 95% CI: 0.06–8.22%, $I^2 = 0.00\%$) trimester (Fig. 5b). Sensitivity analyses of the CZS proportion by continent showed that the estimate for the Americas (4.43%, 95% CI: 3.21–6.07%, $I^2 = 78.10\%$) was lower than that from travel-based studies in Europe (16.13%, 95% CI: 6.88–33.37%, $I^2 = 0.00\%$); Supplementary Fig. B31). The random-effects estimate for Brazil (5.80%, 95% CI: 3.74–8.91%, $I^2 = 85.20\%$) was marginally higher than the overall estimate (Supplementary Fig. B32).

We extracted 23 central estimates of the proportion of pregnancy loss associated with maternal ZIKV infection, which were derived from 17 studies and ranged from 0%[49–51] to 25%[52] (Supplementary Fig. B14). Of these, we included only 14 studies (20 estimates) in the meta-analysis because of our more stringent meta-analysis inclusion criteria (see 'Data analysis' in the Methods). The pooled overall estimates of the proportion of pregnancy loss were 2.48%

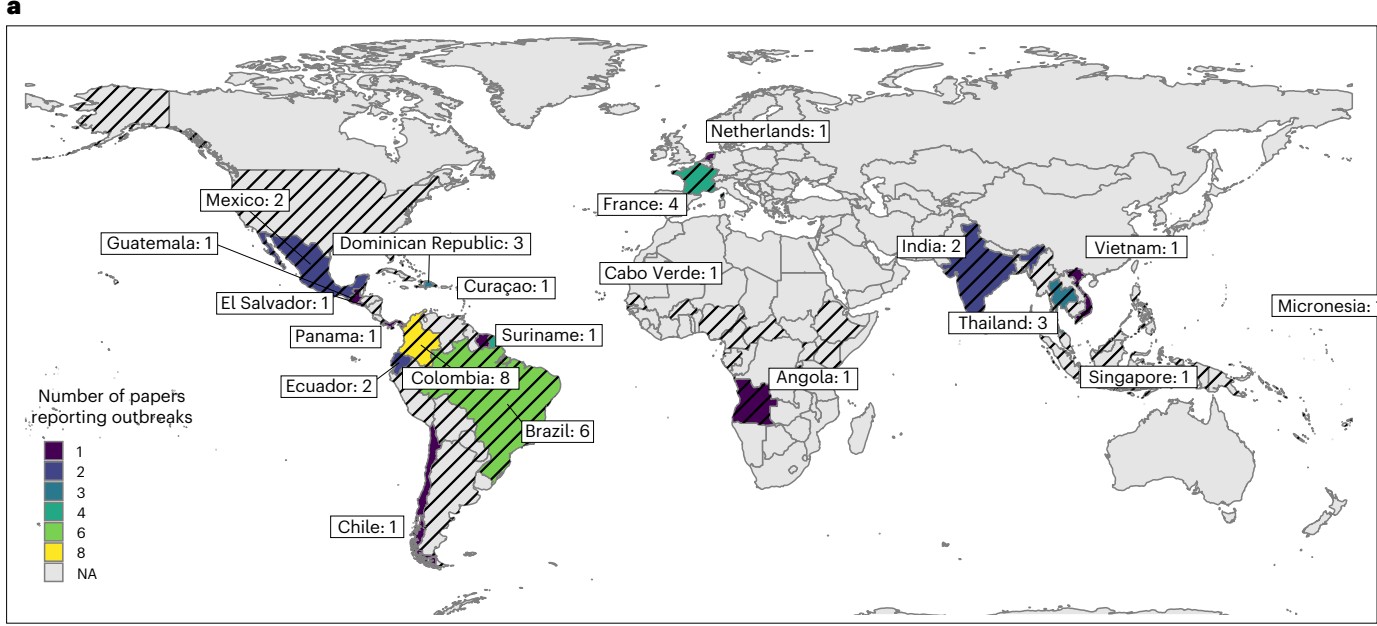

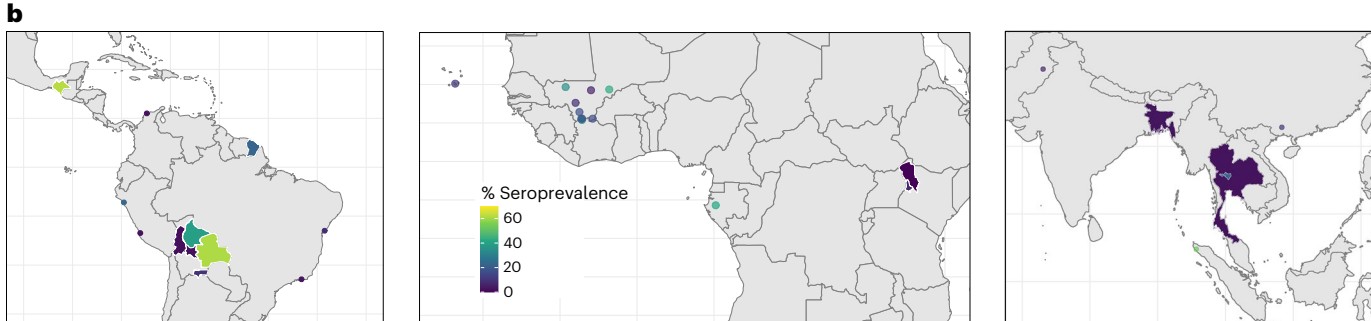

**Fig. 2 | ZIKV outbreak and seroprevalence mapping. a**, Countries with studies reporting ZIKV outbreak information, coloured by number of studies. Outbreaks reported in France and the Netherlands reflect outbreaks in overseas regions. Countries with black diagonal stripes indicate locations where ZIKV transmission has been reported by the WHO[78]. **b**, Geolocated areas or regions with ZIKV seroprevalence studies (using IgG assay, HAI/HI, MIA, NS1 BOB ELISA, IFA,

capture ELISA and neutralization assays) conducted in the general population in the Americas (left), Africa (middle) and Asia (right). Each dot represents a location-specific estimate, while shaded areas indicate estimates at the administrative-unit level (region, province, district or entire country). Base maps adapted from GADM[79]. NA, not applicable.

(95% CI: 1.62–3.78%, $I^2$ = 91.10%) using the random-effects model (Supplementary Fig. B33). When we stratified based on the time of pregnancy loss, the pooled estimate of miscarriage proportion (2.04%, 95% CI: 1.21–3.43%, $I^2$ = 64.70%) was marginally lower than the pooled estimate for general pregnancy loss (3.16%, 95% CI: 1.76–5.61%, $I^2$ = 95.40%); Supplementary Fig. B34).

The CFR estimates (Supplementary Fig. B15 and Table B14) extracted from five studies conducted between 2015 and 2017 in the Dominican Republic[53], the West Indies[54], Brazil[55,56] and Colombia[57] varied from 0.04% reported in a population-based study of 5,161 individuals in the Dominican Republic in 2016 (ref. 53), to 38% in a small hospital-based cohort of 16 persons under investigation in Brazil in 2016 (ref. 56). The proportion of symptomatic individuals (Supplementary Fig. B16 and Table B14) was reported in three studies, yielding ten estimates ranging from 23.2% in pregnant women in French Guiana[58] to 71% (95% CI: 66–76%) in children in the Society Islands (French Polynesia)[59]. Of the three studies, two had sample size information and thus were included in our meta-analysis, which showed a pooled estimate of the proportion of symptomatic cases of 51.20% (95% CI: 38.00–64.23%, $I^2$ = 96.70%; Supplementary Figs. B35 and B36).

## Discussion

In this systematic review, we collated and analysed epidemiological parameters, transmission models and outbreak records of ZIKV to inform future modelling in response to ZIKV outbreaks. One of the clearest findings of this review is the concentration of Zika-related publications within a narrow time window—from the 2015 outbreak in Brazil to the onset of the coronavirus disease 2019 pandemic in 2020. The 2015–2016 ZIKV epidemic in Latin America marked a turning point in ZIKV research: before this, only 6 of the 574 studies included in this review were published, none of which involved modelling. With the emergence of ZIKV as a global health concern, research activity surged, particularly following the outbreak in Brazil in 2015 and the subsequent PHEIC declaration, resulting in a sharp increase in publications (often with a quality assessment score < 50%; Supplementary Fig. B2). Since 2020, however, the volume of new studies has declined despite continued ZIKV circulation, likely due to reduced data availability and shifts in funding priorities after the end of the PHEIC.

Among the 354 estimates characterizing ZIKV exposure through seroprevalence included in this review, there is a marked heterogeneity in geographic representation, with the Americas accounting for the majority of the published seroprevalence estimates. While these

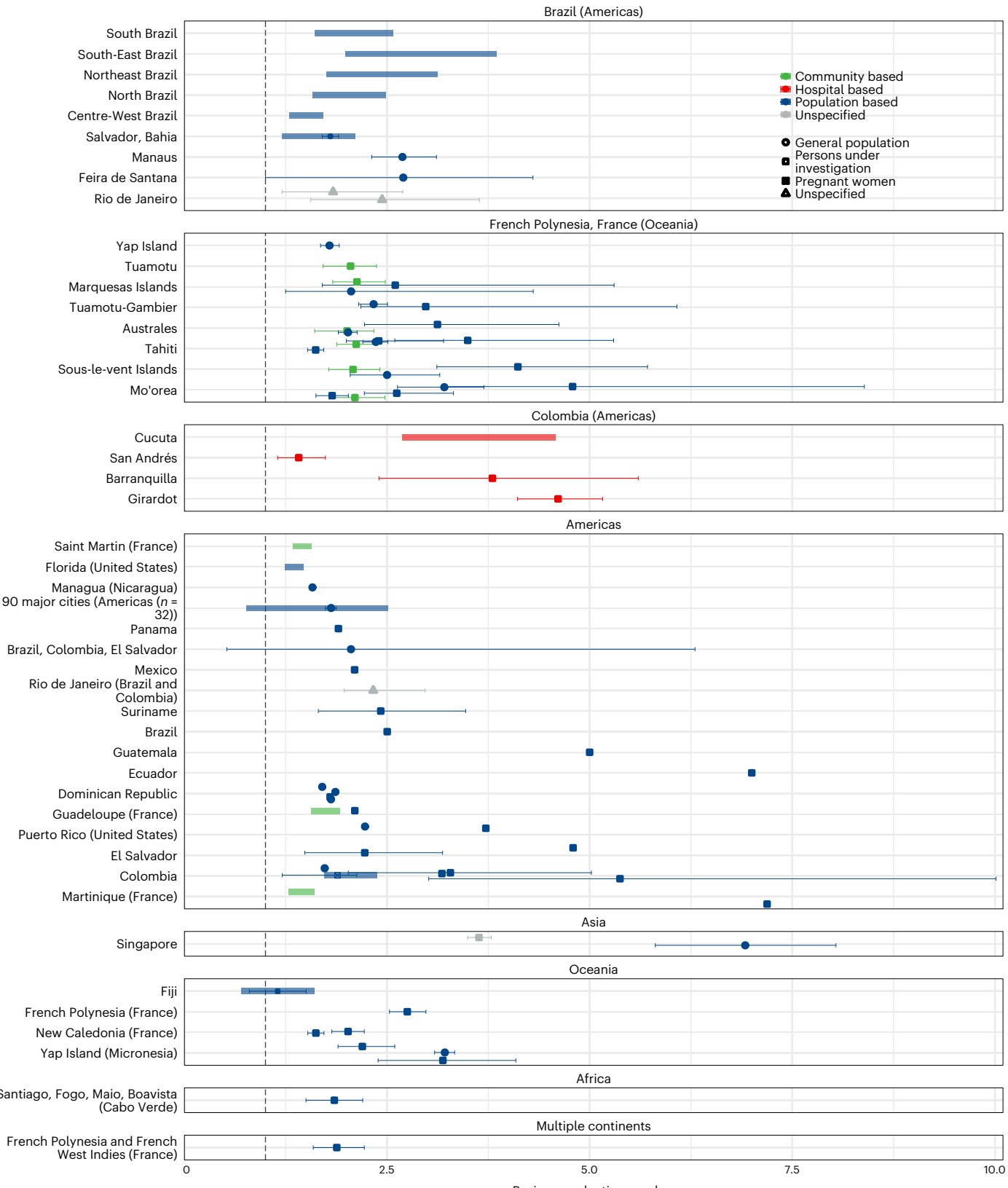

**Fig. 3 | ZIKV basic reproduction number estimates.** $R_0$ estimates by geographic location, type of sample and population group. Sub-national estimates for Brazil, Colombia and French Polynesia are shown in separate panels because of the high number of estimates; all other estimates are shown in the respective continent panels. Overall country estimates for Brazil, Colombia and French Polynesia are shown in the corresponding continent panels. Points are central estimates reported in the published studies (defined as mean, median or unspecified central as estimated in the extracted paper), error bars are 95% confidence or credible intervals, and thicker shaded bars are ranges of central estimates over disaggregated groups. The grey vertical dashed line marks 1. When multiple estimates for the same location were available, the estimates were jittered. The sample sizes for each study shown in the figure are reported in Supplementary Table B12.

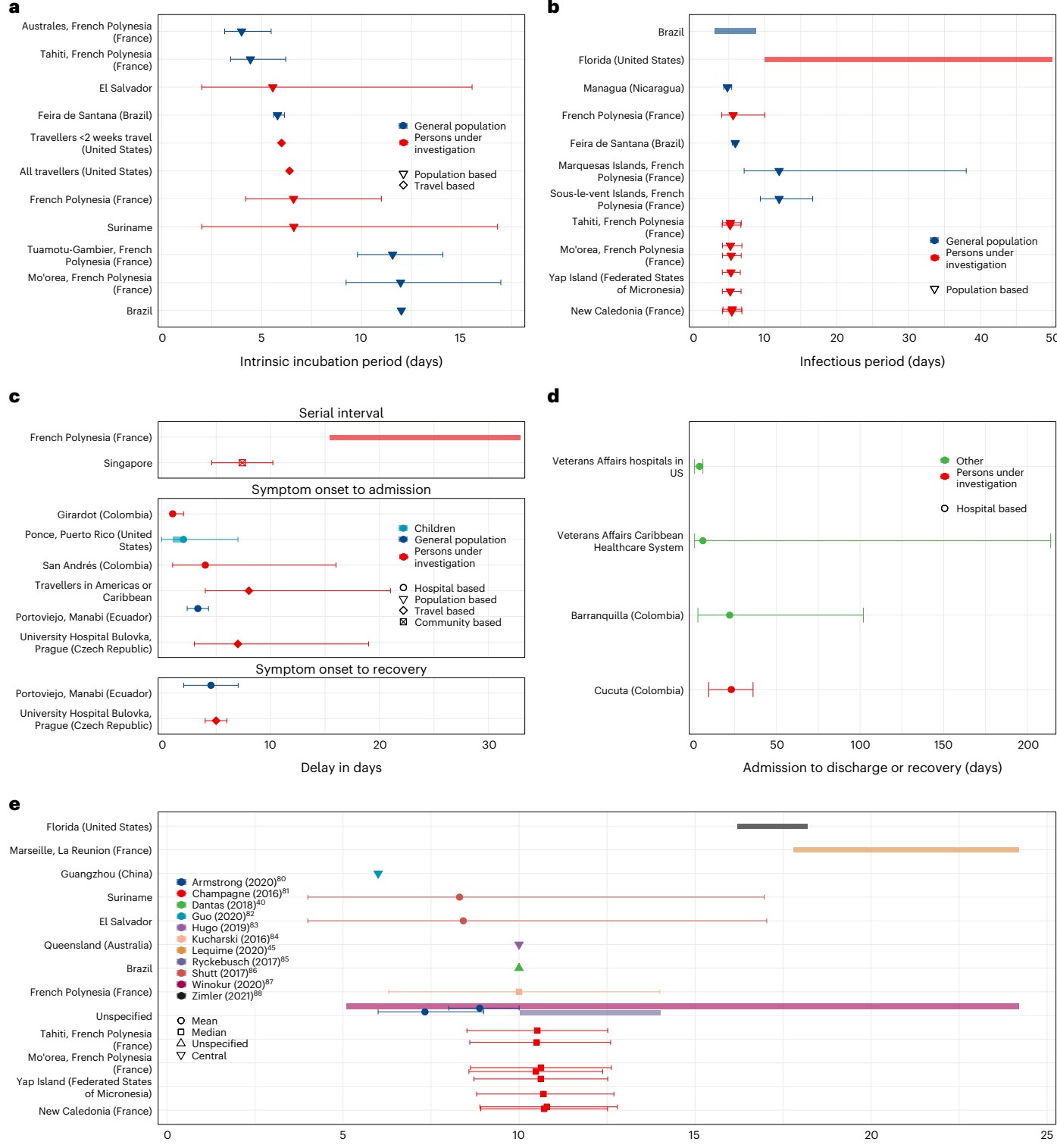

**Fig. 4 | Epidemiological delays in human and mosquitoes. a–e**, Estimates by location and type of sample of Zika intrinsic incubation period (**a**), infectious period (**b**), serial interval, symptom onset to admission and symptom onset to recovery (**c**) and admission to discharge or recovery (**d**). Estimates by location and study of ZIKV extrinsic incubation period in the mosquito (**e**). Points are the central estimates reported in the studies (defined as mean, median or unspecified central estimates as shown in the extracted paper), error bars are 95% confidence or credible intervals, and shaded bars are ranges of central estimates over disaggregated groups. When multiple estimates for the same location were available, the estimates were jittered. The sample sizes for each study shown in the figure are reported in Supplementary Table B9.

statistics reflect the attention that ZIKV received during the 2015–2016 epidemic in the Americas, they also underscore gaps in our understanding of the population-level immunity of populations outside these regions, which is critical to evaluate global outbreak risks now and in the future, and to design clinical trials for future vaccine and therapeutics. Seroprevalence estimates should also be interpreted with caution, as variations in assay sensitivity, specificity and cross-reactivity—especially with other flaviviruses like dengue—could bias the results. Our

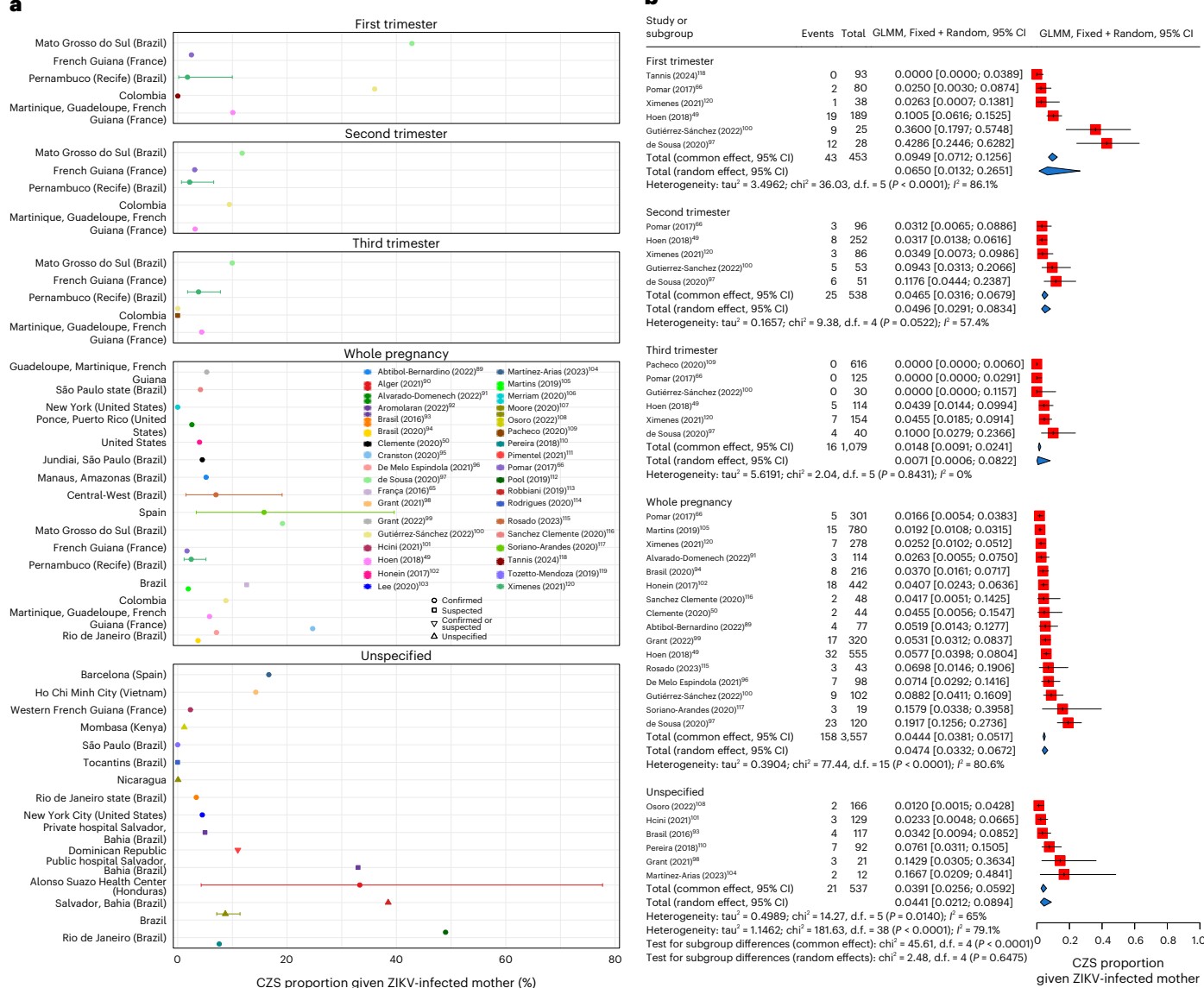

**Fig. 5 | Proportion of CZS. a**, Estimates of reported CZS probabilities by pregnancy stage of the infected mother of an infant with CZS. 'Unspecified' is used when no specific pregnancy stage was reported. Points are central estimates (defined as mean, median or unspecified central values as estimated in the extracted paper), solid lines are confidence or credible intervals, and shaded segments are ranges of central estimates across disaggregated groups. **b**, Meta-analysis of CZS risk stratified by pregnancy stage of the infected mother of an infant with CZS. 'Unspecified' is used when no specific pregnancy stage was reported. We used generalized linear mixed-effect model (GLMM) estimates of CZS proportion. Red squares represent the observed study effect sizes, and the solid black horizontal lines are CIs. Blue diamonds represent pooled common

and random-effect estimates by trimester and overall. Common effect estimates assume that all aggregated data come from a single data-generating process with one common CZS proportion, and overall random-effect estimates allow the CZS proportion to vary by study and give different weights to each study in the overall estimate. The 'Events' column indicates the reported number of CZS cases. The sample sizes for each study shown in the figure are reported in Supplementary Table B16. Statistical analyses were performed using the chi-squared test for heterogeneity, with two-sided significance. Between-study variance was estimated using $tau^2$, and $I^2$ indicates the proportion of total variation due to heterogeneity rather than chance. Figures report the $P$ value for each group, rounding to <0.0001 when values fall below this threshold in the meta-analysis.

analysis has demonstrated that ZIKV is moderately to highly transmissible ($R_0$ typically 1.5–4, up to >7), with both vector-borne and sexual transmission contributing to infection. In combination with a pooled CZS proportion of 4.65% and a pooled pregnancy loss proportion of 2.48%, continued monitoring is essential to track outbreak potential and protect vulnerable populations. Moreover, our meta-analysis on the proportion of ZIKV-infected symptomatic cases suggests that up to half of ZIKV infections may go undetected in surveillance systems that rely on symptomatic reporting (51.20%, 95% CI: 38.00–64.23%, $I^2$ = 96.70%).

Estimates of CZS proportion vary widely due to differences in study design (that is, recruitment based on confirmed infection versus

retrospective serology), population characteristics, case definitions and methods used for the identification of ZIKV infections during pregnancy. Several studies from which we extracted CZS proportion were not specifically designed to assess the CZS proportion in recently infected mothers (that is, studies tested for past exposure by IgG assays rather than recent infections through IgM detection). Our meta-analysis produced an overall estimate of 4.65% (95% CI: 3.38–6.37%, $I^2$ = 77.10%), which is slightly higher than other published estimates[60–63], although the CIs overlap. We also found only one published review that conducted a trimester-specific meta-analysis of the proportion of CZS[64] including only two studies[65,66]. Our meta-analysis of CZS proportion by trimester, based on seven studies, yielded higher estimates than

those published in Gallo et al.[64], with consistently higher estimates in earlier trimesters.

Although we extracted 969 parameters from the literature, it was not possible to perform meta-analyses across most parameters (other than CZS, pregnancy loss probabilities and proportion of symptomatic cases) due to low numbers of comparable estimates, unavailability of sample size information and substantial heterogeneity in study contexts and methodologies. For example, only one estimate of the generation time and two estimates of the serial interval have been published. We also noted substantial variation in the terminology and methods used for estimation in studies reporting attack rates and reproduction number estimates, where basic and effective reproduction numbers were occasionally defined interchangeably, and in studies reporting genomic information, where mutation rates (per generation) and evolutionary or substitution rates (per year) were often conflated.

This study has limitations. We included only peer-reviewed publications in English, which may have led to the exclusion of relevant studies published in other languages and in grey literature or outbreak reports. Studies reporting outbreak dynamics or seroprevalence estimates that did not meet the formal inclusion criteria were excluded[67–71]; hence, the outbreak data reported in this systematic review should not be considered a complete compendium, as several sources of ZIKV outbreaks exist outside peer-reviewed publications, specifically PAHO[72] and national Ministries of Health. Moreover, while we attempted to extract disaggregated parameter estimates where available, for feasibility purposes, we prioritized summarizing data over fully disaggregated information (for example, by age or sex; Supplementary Information). All information is, however, included in our open-source database (also in Supplementary Tables B7–B16) that is accessible via the R package 'epireview' to support future reuse. Finally, data extraction was conducted by a team of 16 reviewers using detailed, standardized guidelines[73]. Although double extractions were conducted at early stages, a residual risk of inconsistency and human error remains.

Despite these limitations, this study has several strengths as it summarizes the information published in 418 studies on 984 epidemiological parameters, 154 models and 127 outbreak records, which provides the most thorough quantitative overview of ZIKV epidemiology and disease risk available to date. As well as documenting key epidemiological parameters needed for outbreak response and preparedness planning, the richness of the data collated in this systematic review allows us to provide global, regional and country-level estimates of the proportion of adverse effects of ZIKV, including CZS, pregnancy loss and symptomatic cases.

In summary, this study presents a dynamic, open-source resource designed to support continuous updates by the research community to provide timely quantitative information on ZIKV epidemiology. By enabling the development of data-driven and evidence-based analytical, modelling and computational tools, this resource will aid mathematical modellers, public health officials and policymakers in outbreak prevention, preparedness, response and disease control planning.

## Methods

We followed the PRISMA guidelines and registered our study protocol with PROSPERO (International Prospective Register of Systematic Reviews, CRD42023393345). The checklist is included in the Supplementary Information.

### Search and screening

We searched PubMed and Web of Science for studies published from database inception to 31 October 2024. Results were imported into Covidence[74] and de-duplicated. Titles, abstracts and then full texts were independently screened by two reviewers, and conflicts were resolved by consensus. The Cohen's Kappa scores for

screening and full-text review are reported in Supplementary Fig. A1. Non-peer-reviewed literature and non-English language studies were excluded. We also excluded papers with co-infection studies, comments, letters, case reports, case studies and those that reported estimates from other papers. Systematic reviews ($n = 66$) were excluded from extraction (Supplementary Information Section A), but peer-reviewed literature identified through backwards citation screening meeting the selection criteria was included. Further details on inclusion and exclusion criteria are provided in the Supplementary Information.

### Data extraction

Of the full texts meeting the inclusion criteria, 16% ($n = 91$) were randomly selected for double extraction to ensure concordance between a team of 16 extractors. After consensus was reached, reviewers independently conducted single extractions on the remaining full texts.

Data were extracted using a custom-made Microsoft Access database (version 2305; Supplementary Tables A4–A6). We collected information on publication details and three categories: transmission models, outbreaks and epidemiological parameters. In the transmission model category, we extracted information on the mathematical or statistical models of transmission used in the included papers. In the outbreak category, we reported outbreak data only when it was newly observed, rather than derived from another publication. Finally, the epidemiological parameters included were basic and effective reproduction numbers, epidemiological delays (the intrinsic incubation period, the extrinsic incubation period, the time between symptom onset to hospitalization or from symptom onset to outcome), CFRs, attack rates, growth rates, evolutionary, mutation and substitution rates, the proportion of symptomatic cases, overdispersion (quantifying heterogeneity in transmission), seroprevalence, proportion of CZS/microcephaly in newborns born to infected mothers, proportion of pregnancy loss in infected pregnant women, relative contributions to transmission from different routes and risk factors related to Zika or CZS. Pregnancy loss proportion was extracted as either the proportion of miscarriage if the study explicitly reported this, or general pregnancy loss, which included both the sum of miscarriage and stillbirth and pregnancy loss with no specified information on the trimester of loss.

We extracted estimates of the basic and effective reproduction numbers ($R_0$ and $R_e$, representing the average number of secondary infections generated by a case in a fully susceptible and partially susceptible population, respectively), the basic vector-borne reproduction number ($R_0^v$, representing the average number of new infections generated via vector-borne transmission in a fully susceptible population) and the basic and effective sexual reproduction numbers ($R_0^h$ and $R_e^h$, representing the average number of new cases generated via sexual transmission in a fully susceptible and partially susceptible population, respectively).

We used a custom quality assessment tool to assess the quality of each study (Supplementary Table A2). Full details on the data extraction process and extracted data are provided in the Supplementary Information.

### Data analysis

The analysis was conducted in R software using the 'orderly2' workflow package (version 1.99.14)[75]. The quality score of each study was calculated as the proportion of 'yes' answers to the total number of applicable questions in the quality assessment tool for that study. Only parameters with a quality score $\geq 50\%$ were included in the analyses presented in the main text. Sensitivity analyses including all studies, regardless of the quality assessment score, are presented in Supplementary Information Section B4.

We conducted meta-analyses for the proportion of symptomatic cases, proportion of CZS and proportion of pregnancy loss among

confirmed ZIKV-infected mothers. For the latter two, we applied more meta-analysis inclusion criteria by only including estimates with at least ten pregnant women with confirmed ZIKV infection and with a study design that did not select for the outcome (CZS or miscarriage), using the 'meta' R package (version 8.2-1)[76]. We used random-effect and fixed-effect models using the logit link function to calculate pooled estimates for both parameters, with 95% CIs and $I^2$ heterogeneity estimates. We report the random-effects estimates in the paper, and the fixed-effect estimate is reported in meta-analysis plots (Fig. 5 and Supplementary Information Section B5). We estimated the pooled CZS and pregnancy loss probabilities by country, continent and sample population type using maximum likelihood to estimate the between-studies variance. Subgroup analyses were considered to assess differences in CZS and pregnancy loss probabilities. Funnel plots assessing publication bias are shown in Supplementary Figs. B29 and B30.

### Reporting summary

Further information on research design is available in the Nature Portfolio Reporting Summary linked to this article.

## Data availability

Data are publicly available from https://github.com/mrc-ide/epireview/.

## Code availability

Code and a vignette to reproduce the analysis are available from https://github.com/mrc-ide/priority-pathogens and https://mrc-ide.github.io/priority-pathogens/articles/pathogen_zika.html. All analyses were performed in R (version 4.5.0) with the following packages: epireview (1.4.5), orderly2 (1.99.14), prioritypathogens (https://github.com/mrc-ide/priority-pathogens/tree/zika_v2 or via Zenodo at https://doi.org/10.5281/zenodo.17201039 (ref. 77)), metafor (4.8-0), meta (8.2-1), estmeansd (1.0.1), mixdist (0.5-5), ggplot2 (4.0.0), ggsci (3.2.0), sf (1.0-21), ragg (1.4.0), ggspatial (1.1.10), ggforce (0.5.0), png (0.1-8), grid (4.5.1), patchwork (1.3.2), gridExtra (2.3), readxl (1.4.5), harrypotter (2.1.1), rnaturalearth (1.1.0), rnaturalearthdata (1.0.0), ggrepel (0.9.6), gt (1.0.0), scales (1.4.0), gpkg (0.0.12), RSQLite (2.4.3), countrycode (1.6.1), ggpattern (1.2.1), tidyverse (2.0.0), stringr (1.5.2), maps (3.4.3) and tidygeocoder (1.0.6). An archived version of the code used for the analysis is available via Zenodo at https://doi.org/10.5281/zenodo.17201039 (ref. 77).

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

## Acknowledgements

The authors acknowledge funding from the Medical Research Council (MRC) Centre for Global Infectious Disease Analysis (MR/X020258/1) funded by the UK MRC and carried out in the frame of the Global Health EDCTP3 Joint Undertaking supported by the European Union. A.C. acknowledges funding from National Institute for Health Research (NIHR) Health Protection Research Unit in Health Analytics & Modelling (NIHR207404), a partnership between the UKHSA, London School of Hygiene & Tropical Medicine and Imperial College of Science, Technology, & Medicine, and was supported by the Academy of Medical Sciences Springboard scheme, funded by the Academy of Medical Sciences, Wellcome Trust, the UK Department for Business, Energy, and Industrial Strategy, the British Heart Foundation and Diabetes UK (SBF005\1044).

K.M., A.V., C.M., T.R., T.M.N., S.B. and S.R.-P. acknowledge funding from data.org (https://data.org/initiatives/epiverse/; Epiverse-TRACE). K.M. acknowledges research funding from the Imperial College President's PhD Scholarship. I.D. and A.V. acknowledge funding from Wellcome Trust (213494/Z/18/Z, 226072/Z/22/Z, 226727/Z/22/Z, and 228185/Z/23/Z). C.M. acknowledges the Schmidt Sciences for research funding (grant code 6–22–63345). T.R., T.M.N. and P.D. acknowledge funding from Community Jameel.

S.B. acknowledges funding from the Wellcome Trust (223120/Z/21/Z). A.-M.H. acknowledges funding from The Federal Ministry of Research, Technology and Space (BMFTR) through the CLIMADEMIC project (funding code 01LN2210A) within the framework of the Strategy Research for Sustainability (FONA). A.-M.H. also acknowledges additional funding from Gavi, the Vaccine Alliance, the Bill and Melinda Gates Foundation (INV-034281), previously (OPP1157270/INV-009125), and from Wellcome Trust (226727/Z/22/Z). G.C.-D. acknowledges funding from the Royal Society. M.K. and P.L. acknowledge funding from the Imperial Policy Forum. N.R.F. acknowledges funding from Wellcome Trust (316633/Z/24/Z) and the United Nations Foundation (G-23279). A.C. was supported by the Academy of Medical Sciences Springboard scheme, funded by the Academy of Medical Sciences, Wellcome Trust, the UK Department for Business, Energy, and Industrial Strategy, the British Heart Foundation and Diabetes UK (SBF005\1044).

The views expressed are those of the authors and not necessarily those of the NIHR, UKHSA or the Department of Health and Social Care. The funders of this study had no role in study design, data collection, data analysis, data interpretation, or writing of the report.

## Author contributions

S.B., S.v.E., N.I.-E. and A.C. conceptualized this systematic review. K.M., A.V., C.M., T.R., S.I.L., H.J.T.U., K.C., Z.M.C.P., E.S.K. and R.M. searched the literature and screened the titles and abstracts. K.M., A.V., C.M., T.R., T.M.N., K.F., A.-M.H., S.I.L., K.C., Z.M.C.P., G.C.-D., E.S.K. and R.M. reviewed all full-text publications. K.M., A.V., C.M., T.R., T.M.N., S.B., D.P.D., P.D., K.F., A.-M.H., S.I.L., S.R.-P., R.J.S., H.J.T.U., R.M. and I.D. extracted the data. K.M. and A.V. did formal analysis of, visualized and validated the data, with supervision from R.M. and I.D. K.M., A.V., T.M.N. and S.B. were responsible for software infrastructure. A.C. acquired funding. A.C., R.M. and I.D. were responsible for project administration. P.D., G.C.-D., R.K.N., D.N., H.J.T.U. and S.v.E. were responsible for training individuals on and accessing Covidence and designing the Access system. A.V. and K.M. accessed and verified the data. K.M., A.V., R.M. and I.D. wrote the original draft of the paper. All authors debated, discussed, edited and approved the final version of the paper. All authors had full access to all the data in the study and had final responsibility for the decision to submit the paper for publication.

## Competing interests

P.D. and R.M. report payment from the WHO for consulting on integrated modelling and MERS-CoV, respectively. H.J.T.U. reports payment from the Moderna Charitable Foundation (paid directly to the institution for an unrelated project). A.C. received personal consulting fees from Munich Re for work unrelated to this project. All other authors declare no competing interests.

## Additional information

**Correspondence and requests for materials** should be addressed to Kelly McCain, Anna Vicco or Ilaria Dorigatti.

Kelly McCain [1,15] ✉, Anna Vicco [1,15] ✉, Christian Morgenstern [1], Thomas Rawson[1], Tristan M. Naidoo[1], Sangeeta Bhatia[1], Dominic P. Dee [1], Patrick Doohan [1], Keith Fraser[1], Anna-Maria Hartner[1,2], Sequoia I. Leuba[1,3], Shazia Ruybal-Pesántez [1,4], Richard J. Sheppard[1], H. Juliette T. Unwin [1,5], Kelly Charniga[1], Zulma M. Cucunubá[6], Gina Cuomo-Dannenburg[1], Natsuko Imai-Eaton [1], Edward S. Knock[1], Adam Kucharski[3], Mantra Kusumgar[1], Paul Liétar[1], Rebecca K. Nash[1], Sabine van Elsland [1], Pathogen Epidemiology Review Group, Nuno R. Faria [1], Anne Cori[1,7], Ruth McCabe[1,16] & Ilaria Dorigatti [1,16] ✉

[1]MRC Centre for Global Infectious Disease Analysis & WHO Collaborating Centre for Infectious Disease Modelling, Jameel Institute, School of Public Health, Imperial College London, London, UK. [2]Centre for Artificial Intelligence in Public Health Research, Robert Koch Institute, Berlin, Germany. [3]London School of Hygiene and Tropical Medicine, London, UK. [4]Instituto de Microbiología, Universidad San Francisco de Quito, Quito, Ecuador. [5]School of Mathematics, University of Bristol, Bristol, UK. [6]Public Health Institute, Pontificia Universidad Javeriana, Bogotá, Colombia. [7]Health Protection Research Unit in Modelling and Economics, London, UK. [15]These authors contributed equally: Kelly McCain, Anna Vicco. [16]These authors jointly supervised this work: Ruth McCabe, Ilaria Dorigatti. ✉e-mail: k.mccain22@imperial.ac.uk; a.vicco21@imperial.ac.uk; i.dorigatti@imperial.ac.uk

## Pathogen Epidemiology Review Group

Aaron Morris[8], Alpha Forna[9], Amy Dighe[1,10], Anna Vicco[1,15], Anna-Maria Hartner[1,2], Anne Cori[1], Arran Hamlet[1], Ben Lambert[8], Bethan Cracknell Daniels[1], Charles Whittaker[1], Christian Morgenstern[1], Cosmo Santoni[1], Cyril Geismar[1], Dariya Nikitin[1], David Jorgensen[1], Dominic P. Dee[1], Edward S. Knock[1], Gina Cuomo-Dannenburg[1], H. Juliette T. Unwin[1,5], Hayley Thompson[11], Ilaria Dorigatti[1,16], Isobel Routledge[12], Jack Wardle[1], Janetta Skarp[1], Joseph Hicks[1], Kanchan Parchani[1], Keith Fraser[1], Kelly Charniga[1], Kelly McCain[1,15], Kieran Drake[1], Lily Geidelberg[1], Lorenzo Cattarino[13], Mantra Kusumgar[1], Mara Kont[1], Marc Baguelin[1], Natsuko Imai-Eaton[1], Pablo N. Perez-Guzman[1], Patrick Doohan[1], Paul Liétar[1], Paula Christen[1], Rebecca Nash[1], Richard Fitzjohn[1], Richard Sheppard[1], Rob Johnson[1], Ruth McCabe[1,16], Sabine van Elsland[1], Sangeeta Bhatia[1], Sequoia I. Leuba[1,3], Shazia Ruybal-Pesántez[1,4], Sreejith Radhakrishnan[14], Thomas Rawson[1], Tristan M. Naidoo[1] & Zulma M. Cucunubá[6]

[8]University of Oxford, Oxford, UK. [9]University of Georgia, Athens, GA, USA. [10]Johns Hopkins University, Baltimore, MD, USA. [11]PATH, Seattle, WA, USA. [12]University of California, San Francisco (UCSF), Oakland, CA, USA. [13]UK Health Security Agency (UKHSA), London, UK. [14]University of Glasgow, Glasgow, UK.

# Reporting Summary

## Statistics

For all statistical analyses, confirm that the following items are present in the figure legend, table legend, main text, or Methods section.

| n/a | Confirmed | |
|---|---|---|
| ☐ | ☒ | The exact sample size (*n*) for each experimental group/condition, given as a discrete number and unit of measurement |
| ☒ | ☐ | A statement on whether measurements were taken from distinct samples or whether the same sample was measured repeatedly |
| ☐ | ☒ | The statistical test(s) used AND whether they are one- or two-sided<br>*Only common tests should be described solely by name; describe more complex techniques in the Methods section.* |
| ☒ | ☐ | A description of all covariates tested |
| ☐ | ☒ | A description of any assumptions or corrections, such as tests of normality and adjustment for multiple comparisons |
| ☐ | ☒ | A full description of the statistical parameters including central tendency (e.g. means) or other basic estimates (e.g. regression coefficient) AND variation (e.g. standard deviation) or associated estimates of uncertainty (e.g. confidence intervals) |
| ☒ | ☐ | For null hypothesis testing, the test statistic (e.g. *F*, *t*, *r*) with confidence intervals, effect sizes, degrees of freedom and *P* value noted<br>*Give P values as exact values whenever suitable.* |
| ☒ | ☐ | For Bayesian analysis, information on the choice of priors and Markov chain Monte Carlo settings |
| ☒ | ☐ | For hierarchical and complex designs, identification of the appropriate level for tests and full reporting of outcomes |
| ☒ | ☐ | Estimates of effect sizes (e.g. Cohen's *d*, Pearson's *r*), indicating how they were calculated |

*Our web collection on statistics for biologists contains articles on many of the points above.*

## Software and code

Policy information about availability of computer code

| Data collection | Screening and full-text review were performed using Covidence software (https://www.covidence.org/). Data extractions were performed with a custom-made Microsoft Access database (version 2305). Data are available at https://github.com/mrc-ide/epireview/. |
|---|---|
| Data analysis | Code to reproduce the published analysis is available at: https://doi.org/10.5281/zenodo.17201039. A vignette with code to reproduce the plots in the main text can be found at: https://mrc-ide.github.io/priority-pathogens/articles/pathogen_zika.html. All analysis was conducted in R (version 4.4.2). Packages used include: epireview (1.4.5), orderly2(1.99.14), prioritypathogens(https://github.com/mrc-ide/priority-pathogens/tree/zika_v2 or found here: https://doi.org/10.5281/zenodo.17201039), metafor(4.8-0), meta(8.2-1), estmeansd(1.0.1), mixdist(0.5-5), ggplot2(4.0.0), ggsci(3.2.0), sf(1.0-21), ragg(1.4.0), ggspatial(1.1.10), ggforce(0.5.0), png(0.1-8), grid(4.5.1), patchwork(1.3.2), gridExtra(2.3), readxl(1.4.5), harrypotter(2.1.1), rnaturalearth(1.1.0), rnaturalearthdata(1.0.0), ggrepel(0.9.6), gt(1.0.0), scales(1.4.0), gpkg(0.0.12), RSQLite(2.4.3), countrycode(1.6.1), ggpattern(1.2.1), tidyverse(2.0.0), stringr(1.5.2), maps(3.4.3), tidygeocoder(1.0.6). |

For manuscripts utilizing custom algorithms or software that are central to the research but not yet described in published literature, software must be made available to editors and reviewers. We strongly encourage code deposition in a community repository (e.g. GitHub). See the Nature Portfolio guidelines for submitting code & software for further information.

## Data

Policy information about availability of data

All manuscripts must include a data availability statement. This statement should provide the following information, where applicable:
- Accession codes, unique identifiers, or web links for publicly available datasets
- A description of any restrictions on data availability
- For clinical datasets or third party data, please ensure that the statement adheres to our policy

Data extractions were performed with a custom-made Microsoft Access database (version 2305). Data are available at https://github.com/mrc-ide/epireview/. There are no restrictions on data availability.

## Research involving human participants, their data, or biological material

Policy information about studies with human participants or human data. See also policy information about sex, gender (identity/presentation), and sexual orientation and race, ethnicity and racism.

| | |
|---|---|
| Reporting on sex and gender | We assessed the probability of Zika congenital syndrome in pregnant women, as defined by the included papers, using secondary data extracted from the included papers. |
| Reporting on race, ethnicity, or other socially relevant groupings | We did not extract information on race, ethnicity, or other socially relevant groupings. |
| Population characteristics | We extracted information about age, sex/gender (as defined in the papers), Zika positivity based on diagnostic and serological tests, and population type (e.g. children, pregnant women, hospitalized patients, population-based participants), and location (country and admin 1 unit). |
| Recruitment | We used secondary data extracted from papers included in our review. In our meta-analysis, we evaluated potential bias using funnel plots. |
| Ethics oversight | No ethics approval was required because we used only previously published secondary data. |

Note that full information on the approval of the study protocol must also be provided in the manuscript.

# Field-specific reporting

Please select the one below that is the best fit for your research. If you are not sure, read the appropriate sections before making your selection.

☒ Life sciences  ☐ Behavioural & social sciences  ☐ Ecological, evolutionary & environmental sciences

For a reference copy of the document with all sections, see nature.com/documents/nr-reporting-summary-flat.pdf

# Life sciences study design

All studies must disclose on these points even when the disclosure is negative.

| | |
|---|---|
| Sample size | No sample size calculation was performed because we used only previously published secondary data. |
| Data exclusions | In the main analysis, data from papers with a quality assessment score below 50% were excluded. For the meta-analysis, we used stringent criteria for inclusion, specified in the Methods section and the Extended Methods section in the Appendix. Specifically, we conducted meta-analyses for the proportion of symptomatic cases, probability of CZS and probability of pregnancy loss among confirmed ZIKV-infected mothers. For the latter two, we only included estimates with at least 10 pregnant women with confirmed ZIKV infection and with a study design that did not select for the outcome (CZS or miscarriage), using the meta R package. |
| Replication | We did not conduct any experiments, but all analysis can be reproduced using the code at: https://doi.org/10.5281/zenodo.17201039. |
| Randomization | This is not relevant to our study as we only used previously published secondary data. |
| Blinding | This is not relevant to our study as we only used previously published secondary data. |

# Reporting for specific materials, systems and methods

We require information from authors about some types of materials, experimental systems and methods used in many studies. Here, indicate whether each material, system or method listed is relevant to your study. If you are not sure if a list item applies to your research, read the appropriate section before selecting a response.

## Materials & experimental systems

| n/a | Involved in the study |
|-----|----------------------|
| ☒ ☐ | Antibodies |
| ☒ ☐ | Eukaryotic cell lines |
| ☒ ☐ | Palaeontology and archaeology |
| ☒ ☐ | Animals and other organisms |
| ☒ ☐ | Clinical data |
| ☒ ☐ | Dual use research of concern |
| ☒ ☐ | Plants |

## Methods

| n/a | Involved in the study |
|-----|----------------------|
| ☒ ☐ | ChIP-seq |
| ☒ ☐ | Flow cytometry |
| ☒ ☐ | MRI-based neuroimaging |

## Plants

| Seed stocks | NA |
|-------------|-----|
| Novel plant genotypes | NA |
| Authentication | NA |

