## [Peer Review File · Nature Health]

A systematic review and meta-analysis of Zika Virus epidemiology

Corresponding Author: Ms Kelly McCain

Version 0:

Reviewer comments:

Reviewer #1

(Remarks to the Author)

A. B. C. D. E.

The article provides a comprehensive systematic review of Zika virus (ZIKV) disease mathematical models and epidemiological parameters, covering 574 studies, 159 outbreaks, 229 models, and 1,334 parameters. It includes meta-analyses of key delays such as the incubation period and serial interval, along with analyses of birth outcome severity, including the probability of congenital Zika syndrome (CZS) and pregnancy loss. The review also estimates the basic reproduction number of ZIKV. A notable contribution is the introduction of an open-access R package, *epireview*, designed to centralize and share data. The work's outstanding features include its extensive scope and scale, as well as the integration of modeling and epidemiological data. The manuscript is methodologically sound and contains no flaws that would prevent its publication. The authors followed PRISMA guidelines, registered the review with PROSPERO, and employed rigorous quality control measures, such as double-screening and double data extraction. They also used appropriate statistical models, applying both random and fixed effects where suitable. The meta-analyses appropriately use random-effects models to account for heterogeneity, with I^2 statistics reported to measure variability across studies. Confidence intervals and error bars are clearly shown in the figures, improving interpretability. Overall, the statistical methods used are appropriate and well-described, enhancing the robustness and transparency of the analysis.

F.

I have only minor edits/suggestions

1. Figure B26: It would be helpful to indicate the reference line for $R_0=1$. Since the vertical dashed line represents this threshold, it should be clearly labeled and mentioned in the figure caption for clarity.
2. Figure B6 (Panel B): Please clarify whether this depicts the vector-borne reproduction number or the sexual (human-to-human) reproduction number. It would be beneficial to explain the distinctions between the different reproduction numbers used throughout the manuscript, specifically R_0 , R_0^h , R_0^v , and R_e^h .
3. Geographic Summary: Including a summary of location-specific parameter estimates (e.g., for the Americas, Africa, and Asia) could provide valuable context for readers. For instance, estimating mean R_0 values by region--Americas, Africa, and Asia-- would help illustrate how transmission dynamics vary geographically.
4. Line 224. The phrase "under our stringent inclusion criteria" is unclear. Are the authors referring to criteria beyond those listed in Table A1? It seems that the 53 estimates from 36 studies mentioned in line 221 were already subject to the inclusion/exclusion criteria outlined in Table A1. If additional or more restrictive criteria were applied at this stage, it would be helpful to specify them. The same clarification is needed for the statement on line 236.
5. The term GLMM should be defined in the captions of Figures B31, B32, and B33 for clarity. Providing a brief explanation (e.g., "generalized linear mixed model") would help readers unfamiliar with the acronym understand the methodology.
6. The manuscript refers to the *epireview* package as the source for data availability, but the linked GitHub page primarily serves as an

(Remarks on code availability)

6. The manuscript refers to the `epireview` package as the source for data availability, but the linked GitHub page primarily serves as an introduction to the package itself. From my understanding, the authors have developed `epireview` as a valuable tool for conducting meta-analyses in R, and its utility is evident. However, to enhance transparency and reproducibility, it would be helpful to include a dedicated folder within the GitHub repository that contains the Zika-specific datasets and the scripts used to generate the figures presented in the manuscript. Providing this additional material would make it easier for other researchers to reproduce the results and apply similar methods to related research questions.

Reviewer #3

(Remarks to the Author)

Drs McCain, Vicco and colleagues report a systematic review with meta-analysis on several aspects of the epidemiology of Zika virus disease. This is a huge undertaking, with 28,000 studies screened and more than 500 included, making the findings significant if not original. The methods are rigorous and up to the state of the art in systematic reviews, including a pre-registration of the protocol, an evaluation of the risk of bias and a correct handling of uncertainty. I also commend the authors for making their data available in an R package. The results are clear and well-presented, and support the conclusions. The figures are high-quality. Overall, I only have a few suggestions about formulation, and my assessment is that this paper should be accepted for publication.

Minor points

- Overall: Just personal preference, but I would suggest rounding most estimates to 1 or 2 significant digits (I feel that 2.5% or even 2% is more adequate than 2.48%, especially in a context of large heterogeneity and uncertainty).
- Abstract: I find the formulation "a pooled total random effect of CZS probability" a bit awkward. I would suggest something like "Using a random-effects model, we estimated the probabilities of congenital Zika syndrome at 4%, pregnancy loss at 2%, and symptomatic cases at 51%", but the authors should feel free to adapt. On the same sentence, I don't see why CSZ and pregnancy loss are considered probabilities while symptomatic cases are a proportion, I would use the same term. I would also add uncertainty intervals. On the following sentence, I would suggest adding a range of seroprevalence estimates.
- Line 75: "while many ZIKV infections are asymptomatic" (especially since you conclude to 51% symptomatic)
- Methods: I would suggest adding some details about the inclusion and exclusion criteria, which are implicit. It would also be good to define the categories that are used in the results (outbreak records, models, parameters).
- Line 177: "almost one third"
- Line 197: I'm confused by the distinction between next generation matrix method and compartmental models, as the former is a method used in the context of the latter.
- Results: The I-squared are not reported consistently.

(Remarks on code availability)

Open Access This Peer Review File is licensed under a Creative Commons Attribution 4.0 International License, which permits use, sharing, adaptation, distribution and reproduction in any medium or format, as long as you give appropriate credit to the original author(s) and the source, provide a link to the Creative Commons license, and indicate if changes were

made.

Reviewers' Comments:

We thank the reviewers for taking the time to review our manuscript. Their insightful comments have improved the quality of our work.

Reviewer #1 (Remarks to the Author):

A. B. C.D.E.

The article provides a comprehensive systematic review of Zika virus (ZIKV) disease mathematical models and epidemiological parameters, covering 574 studies, 159 outbreaks, 229 models, and 1,334 parameters. It includes meta-analyses of key delays such as the incubation period and serial interval, along with analyses of birth outcome severity, including the probability of congenital Zika syndrome (CZS) and pregnancy loss. The review also estimates the basic reproduction number of ZIKV. A notable contribution is the introduction of an open-access R package, *epireview*, designed to centralize and share data. The work's outstanding features include its extensive scope and scale, as well as the integration of modeling and epidemiological data. The manuscript is methodologically sound and contains no flaws that would prevent its publication. The authors followed PRISMA guidelines, registered the review with PROSPERO, and employed rigorous quality control measures, such as double-screening and double data extraction. They also used appropriate statistical models, applying both random and fixed effects where suitable. The meta-analyses appropriately use random-effects models to account for heterogeneity, with I^2 statistics reported to measure variability across studies. Confidence intervals and error bars are clearly shown in the figures, improving interpretability. Overall, the statistical methods used are appropriate and well-described, enhancing the robustness and transparency of the analysis.

F. I have only minor edits/suggestions

1. Figure B26: It would be helpful to indicate the reference line for $R_0=1$. Since the vertical dashed line represents this threshold, it should be clearly labeled and mentioned in the figure caption for clarity.

We thank the reviewer for this helpful comment. We have now added a clear sentence explaining that the vertical dotted line in each figure of basic or effective reproduction numbers refers to the reference value 1. We have also expanded the captions to include information about what the points, lines, and shaded bars represent, in line with the Figure 3 caption.

Please see Figure 3 (pp 14) as well as Figures B.5, B.6, B.7, B.8 and B.26 of the Supplementary Information: “The grey vertical dotted line marks 1.”

2. Figure B6 (Panel B): Please clarify whether this depicts the vector-borne reproduction number or the sexual (human-to-human) reproduction number. It would be beneficial to explain the distinctions between the different reproduction numbers used throughout the manuscript, specifically R_0 , R_0^h , R_0^v , and R_e^h .

Thank you for this helpful comment. We have clarified the caption of B6 (pp 18 of Supplementary Information), explaining that panel B shows the vector-borne basic reproduction number estimates. We also explain the difference between each of the reproduction numbers in the Methods section (lines 295-301), but we have repeated these definitions of Basic reproduction numbers and Effective reproduction numbers in the relevant sections of the Supplementary Information for clarity.

“Line 295-301: We extracted estimates of the basic and effective reproduction numbers (R_0 and R_e , representing the average number of secondary infections generated by a case in a fully susceptible and partially susceptible population, respectively), the basic vector-borne reproduction number (R_0^v , representing the average number of new infections generated via vector-borne transmission in a fully susceptible population), and the basic and effective sexual reproduction number (R_0^h and R_e^h , representing the average number of new cases generated via sexual transmission in a fully susceptible and partially susceptible population, respectively).”

“Figure B.6: **Basic sexual reproduction numbers (R_0^h) (A), and basic vector-borne reproduction numbers (R_0^v) (B).** Points are central estimates reported in the published studies, error bars are 95% confidence or credible intervals, thicker shaded bars are ranges of central estimates over disaggregated groups. The grey vertical dotted line marks 1. When multiple estimates for the same location were available, the estimates were jittered. Note: the plot of overall basic reproduction numbers from all studies (regardless of QA score) is Figure B.26.”

3. Geographic Summary: Including a summary of location-specific parameter estimates (e.g., for the Americas, Africa, and Asia) could provide valuable context for readers. For instance, estimating mean R_0 values by region--Americas, Africa, and Asia-- would help illustrate how transmission dynamics vary geographically.

We thank the reviewer for this thoughtful suggestion. We agree that providing geographic context is important. To this end, we present a continent-level meta-analysis of microcephaly/ZIKV risk in the Supplementary Information (Appendix: Figure B31, pp 39), which we now highlight more clearly in the main text (line 169-173).

“Sensitivity analyses of the CZS proportion by continent showed that the estimate for the Americas (4.43% [95%CI: 3.21%-6.07%, $I^2= 78.10\%$]) was lower than that from travel-based studies in Europe (16.13% [95%CI: 6.88%-33.37%, $I^2= 0.00\%$]) (Appendix: Figure B31).”

For other parameters such as R_0 and seroprevalence, we have not provided continent-level summaries, as these are highly context-specific and vary substantially within regions. Additionally, different methods used to produce these estimates often yield inappropriate syntheses. Therefore, we believe that averaging across entire continents could be misleading, but all of the information is available in our R package `epireview` for researchers to conduct their own analysis of this nature should they wish.

4. Line 224. The phrase “under our stringent inclusion criteria” is unclear. Are the authors referring to criteria beyond those listed in Table A1? It seems that the 53 estimates from 36 studies mentioned in line 221 were already subject to the inclusion/exclusion criteria outlined in Table A1. If additional or more restrictive criteria were applied at this stage, it would be helpful to specify them. The same clarification is needed for the statement on line 236.

We thank the reviewer for pointing this out and agree that the phrasing was not sufficiently clear. In the Results section, the statements at previous lines 224 and 236 refer to the more stringent inclusion criteria applied specifically for the meta-analysis only, which are described in detail in the Methods section in lines 311-314. We have now added explicit references to the Methods section in both places (lines 164 and 177) to clarify this.

5. The term GLMM should be defined in the captions of Figures B31, B32, and B33 for clarity. Providing a brief explanation (e.g., “generalized linear mixed model”) would help readers unfamiliar with the acronym understand the methodology.

Thank you for this helpful suggestion. We have updated the legends of all figures with meta-analyses to include a brief explanation of the methods used for meta-analysis. See an example of the added text for Figure 5 (lines 438-448) below:

“We used generalised linear mixed-effect model (GLMM) estimates of CZS proportion. Red squares represent the observed study effect sizes, and the solid black horizontal lines are confidence intervals. Blue diamonds represent pooled common and random effect estimates by trimester and overall. Common effect estimates assume that all aggregated data come from a single data-generating process with one common CZS proportion, and overall random effect estimates allow the CZS proportion to vary by study and give different weights to each study in the overall estimate. The “Events” column indicates the reported number of CZS cases. The vertical dashed line is the

overall pooled estimate.”

Reviewer #1 (Remarks on code availability):

6. The manuscript refers to the epireview package as the source for data availability, but the linked GitHub page primarily serves as an introduction to the package itself. From my understanding, the authors have developed epireview as a valuable tool for conducting meta-analyses in R, and its utility is evident. However, to enhance transparency and reproducibility, it would be helpful to include a dedicated folder within the GitHub repository that contains the Zika-specific datasets and the scripts used to generate the figures presented in the manuscript. Providing this additional material would make it easier for other researchers to reproduce the results and apply similar methods to related research questions.

Thank you very much for the helpful suggestion. We have now created a vignette that reproduces the figures from the manuscript, which should make it easier for others to follow the workflow step by step. Please find it at https://mrc-ide.github.io/priority-pathogens/articles/pathogen_zika.html. We have also added a link to a Zenodo repository where the entire code for the analysis is now available (<https://doi.org/10.5281/zenodo.17201039>). We are confident that these additions will provide clearer guidance for researchers who wish to reproduce the results.

Reviewer #3 (Remarks to the Author):

Drs McCain, Vicco and colleagues report a systematic review with meta-analysis on several aspects of the epidemiology of Zika virus disease. This is a huge undertaking, with 28,000 studies screened and more than 500 included, making the findings significant if not original. The methods are rigorous and up to the state of the art in systematic reviews, including a pre-registration of the protocol, an evaluation of the risk of bias and a correct handling of uncertainty. I also commend the authors for making their data available in an R package. The results are clear and well-presented, and support the conclusions. The figures are high-quality. Overall, I only have a few suggestions about formulation, and my assessment is that this paper should be accepted for publication.

Minor points

- Overall: Just personal preference, but I would suggest rounding most estimates to 1 or 2 significant digits (I feel that 2.5% or even 2% is more adequate than 2.48%, especially in a context of large heterogeneity and uncertainty).

We thank the reviewer for this suggestion. For parameters extracted directly from previously published studies, we have reported the values exactly as presented in the original publications. We chose to retain the original values rather than rounding them, so that readers have access to the precise data; rounding by us would prevent readers from directly recovering the exact original values which is one of the main goals of our work.

- **Abstract:** I find the formulation “a pooled total random effect of CZS probability” a bit awkward. I would suggest something like “Using a random-effects model, we estimated the probabilities of congenital Zika syndrome at 4%, pregnancy loss at 2%, and symptomatic cases at 51%”, but the authors should feel free to adapt. On the same sentence, I don’t see why CSZ and pregnancy loss are considered probabilities while symptomatic cases are a **proportion**, I would use the same term. I would also add **uncertainty** intervals.

We thank the reviewer for these suggestions. We have changed all references to CZS and pregnancy loss probabilities to CZS proportion and pregnancy loss proportion for consistency. We have updated this in the main text, supplementary information text, and in Figure 5, as well as all CZS and pregnancy loss figures in the supplement.

We have also added uncertainty intervals to the random effects estimates and have re-worded the sentence to improve clarity in the abstract in lines 41-43:

“Using random-effects models, we estimated proportions of CZS (4.65% [95%CI: 3.38%-6.67%]), pregnancy loss (2.48% [95%CI: 1.62%-3.78%]), and symptomatic cases (51.20% [95%CI: 38.00%-64.23%]).”

On the following sentence, I would suggest adding a **range of seroprevalence** estimates.

We thank the reviewer for the suggestion. In our review, seroprevalence estimates vary widely because we extracted data from diverse populations, including highly exposed groups. As a result, values range from near 0% to 100%. Given this extreme variability, we felt that reporting a specific range in the abstract could be misleading, so we have chosen to describe the results qualitatively instead.

- Line 75: “while **many** ZIKV infections are asymptomatic” (especially since you conclude to 51% symptomatic)

We thank the reviewer for this comment, it has now been updated (line 70).

“Whilst many ZIKV infections are asymptomatic..”

- **Methods:** I would suggest adding some details about the inclusion and exclusion

criteria, which are implicit. It would also be good to define the categories that are used in the results (outbreak records, models, parameters).

We thank the reviewer for this helpful suggestion. The full inclusion and exclusion criteria are provided in the Supplementary Information (pp 6), but we have now also added a summary of these criteria in the Method section (lines 270-272), together with a reference to the SI for further detail. In addition, we have explicitly defined the different categories (outbreak records, models, parameters, etc.) in lines 281-285 to make the structure of the Results clearer.

- Line 177: “**almost** one third”

We thank the reviewer for this suggestion. We have now updated this in the main text (lines 116).

“Almost one-third of these estimates were..”

- Line 197: I’m confused by the distinction between next generation matrix method and compartmental models, as the former is a method used in the context of the latter.

Thank you for pointing this out. We have updated the description to clarify that the next generation matrix method is a subgroup of compartmental models (lines 135-137):

“Compartmental models were used most frequently (n=45), and among these, the next-generation matrix method was used for 27 estimates.”

- **Results:** The I-squared are not reported consistently.

We have now updated the manuscript to consistently report the I-squared values throughout (lines 176-180, 185-188 and 230).